# Protective Effect of *Vitis labrusca* Leaves Extract on Cardiovascular Dysfunction through HMGB1-TLR4-NFκB Signaling in Spontaneously Hypertensive Rats

**DOI:** 10.3390/nu12103096

**Published:** 2020-10-11

**Authors:** Hye Yoom Kim, Mi Hyeon Hong, Jung Joo Yoon, Dae Sung Kim, Se Won Na, Youn Jae Jang, Yun Jung Lee, Dae Gill Kang, Ho Sub Lee

**Affiliations:** 1College of Oriental Medicine and Professional Graduate School of Oriental Medicine, Wonkwang University, Iksan 54538, Korea; hyeyoomc@naver.com (H.Y.K.); mihyeon123@naver.com (M.H.H.); mora16@naver.com (J.J.Y.); sewon3066@naver.com (S.W.N.); j8626@naver.com (Y.J.J.); shrons@wku.ac.kr (Y.J.L.); 2Hanbang Cardio-Renal Research Center & Professional Graduate School of Oriental Medicine, Wonkwang University, Iksan 54538, Korea; 3Hanpoong Pharm and Foods Co., Ltd., Wanju 55316, Korea; kimezz@naver.com

**Keywords:** *Vitis labrusca* leaves (HP1), hypertension, cardiovascular remodeling, HMGB1-TLR4-NFκB signaling

## Abstract

The *Vitis labrusca* is a grapevine that has antioxidant, neuroprotective, hepatoprotective, and anticarcinogenic activity. However, the effect of *Vitis labrusca* leaves on the cardiovascular system is yet to be ascertained. The present study was designed to investigate the effects of *Vitis labrusca* leaves extract (HP1) on cardiovascular remodeling in spontaneously hypertensive rats. Experiments were performed in rats and were randomly divided into the following groups: Wistar Kyoto rat (WKY), normal control group; spontaneously hypertensive rats (SHR), negative control group; SHR + Losa, positive control group (losartan, 10 mg/kg/daily, AT_1_ receptor blocker) and SHR + HP1 (100 mg/kg/daily). HP1 was orally administered daily for 4 weeks. The HP1 treatment significantly improved blood pressure, electrocardiographic parameters, and echocardiogram parameters compared to hypertensive rats. Additionally, the left ventricular (LV) remodeling and LV dysfunction were significantly improved in HP1-treated hypertensive rats. Furthermore, an increase in fibrotic area has been observed in hypertensive rats compared with WKY. However, administration of HP1 significantly attenuated cardiac fibrosis in hypertensive rats. Moreover, HP1 suppressed the expression of high mobility group box 1 (HMGB1), toll-like receptor 4 (TLR4), myeloid differentiation primary response 88 (MyD88), nuclear factor kappa-light-chain-enhancer of activated B cells (NFκB), tumor necrosis factor alpha (TNF-α), interleukin-6 (IL-6), receptor for advanced glycation end products (RAGE), and extracellular signal-regulated kinases (ERK1/2) induced by hypertensive rats, resulting in improved vascular remodeling. Therefore, these results suggest that HP1 can improve the cardiovascular remodeling in hypertensive rats, and the mechanisms may be related to the suppressive effect of HP1 on HMGB1-TLR4-NFκB signaling in the cardiovascular system. Thus, the protective role of the traditional herbal medicine HP1 may provide new insights into the development of therapeutic drugs on the development of hypertensive cardiovascular dysfunction.

## 1. Introduction

Hypertension-related death is a major cause in modern society and is often associated with cardiovascular damage. Hypertension with left ventricular (LV) hypertrophy and heart failure is a major independent risk factor for cardiovascular-related exchange rates and mortality in the aging population [1]. Furthermore, hypertension is attributed to structural and functional changes in the heart, such as LV hypertrophy, left atrial enlargement, and impairment of LV systolic and diastolic function [2]. LV hypertrophy is a pathological change that causes myocardial fibrosis due to increased collagen content, which causes heart failure [3]. Inflammation is an important risk factor for end-stage organ damage [4,5,6,7] and has been implicated in hypertension [8,9]. Inflammatory cytokines play an essential role in the pathological physiology of cardiovascular disease [6,7,8,9,10]. Damage to the barrier function of endothelial cells further increases the inflammatory response [11] that causes cardiovascular dysfunction [12]. It suggests that the inflammatory mechanism is an important participant in the pathology of hypertension. For the treatment of hypertension, long-term use of anti-hypertensive medication to treat high blood pressure can cause serious drug-resistance or side-effects. Therefore, natural substances used in traditional medicine are of interest in drug development because they have fewer side effects.

Campbell Early grape (*Vitis labrusca* cultivar) is a species of grapevine belonging to Vitaceae. *Vitis labrusca* are today more popular in Asia than in the United States, with two-thirds of the grape varieties cultivated in South Korea being Campbell Early [13,14]. According to previous reports, *Vitis labrusca* is known to have anti-oxidant, anti-carcinogenic, neuroprotective, hepatoprotective, and cardioprotective properties [15,16,17]. However, no previous report has described the protective effect of *Vitis labrusca* leaves on cardiovascular dysfunction in hypertensive models. Thus, the present study was designed to investigate the effects of *Vitis labrusca* leaves extract (HP1) on cardiovascular remodeling in hypertensive rats. The purpose of this study is to assess whether HP1 improves cardiovascular dysfunction by preventing heart fibrosis and LV hypertrophy in hypertensive rats. Furthermore, we investigated whether HP1 mediates the inhibition of anti-inflammation activity in the aspect of attenuating myocardial remodeling.

## 2. Materials and Methods

### 2.1. Preparation of Vitis labrusca Leaf (HP1) Extract

The *Vitis labrusca* leaves extract (HP1 or HP-01) was provided by Hanpoong Pharm and Foods Co., Ltd. (Jeonju, Korea). The vegetal material of Campbell Early grape (Vitis labruscana Bailey) leaves was collected from Hwaseong, Gyeonggi-do in June and July 2019 and extracted by the Hanpoong Pharm and Foods Co., Ltd. (Wanju, Korea). Campbell Early grape (Vitis labruscana Bailey) leaves were dried (5.0 kg) and refluxed with 50% Ethanol (*v*/*v*) and 3% acetic acid (*v*/*v*) at 85 °C, for 3 h. After filtration, the residue was re-extracted as described above, two additional times. The combined filtrates were then evaporated and vacuum dried (HP1, 544.8 g).

### 2.2. High-Performance Liquid Chromatography (HPLC) Analysis Method2

HPLC quantitative analyses were carried out using a Shimadzu LC-2030C 3D HPLC instrument and a photodiode-array (PDA) detector (Shimadzu, Japan). Analysis of Quercetin-3-O-glucuronide was performed using a Waters Xbridge Shield RP18 (150 × 4.6 mm I.D; 3.5 μm) (Waters Corporation, Milford, MA, USA) column.

### 2.3. Experimental Animals and Blood Pressure

Experiments were performed in the male spontaneously hypertensive rats (SHRs, 7 weeks) and Wistar-Kyoto rats (WKY, 7 weeks) used in this study were purchased from SLC Inc. (Shizuoka, Japan) in individual cages in a room kept temperature-controlled environment with 12 to 12 h light and dark cycle. After stabilizing for a week, noninvasive blood pressure was measured using tail-cuff CODA™ High Throughput System (Kent Scientific Corporation, Torrington, CT, USA) in all animal groups. Experiments were performed in rats were divided into the following groups: WKY (*n* = 10), normal control group; SHR (*n* = 12), negative control group; SHR + losartan, positive control group (*n* = 12, 10 mg/kg/daily, AT_1_ receptor blocker) and SHR + HP1 (*n* = 12, 100 mg/kg/daily). Based on the initial study results, we conducted a 100 mg/kg/daily concentration [18]. HP1 was orally administered daily for 4 weeks. Experiments described in this study were carried out following the principle of guidance for animal protection and use by the American Physiological Society. All experimental animal protocols were approved by the Institutional Animal Care and Use Committee of the Wonkwang University (WKU20-26).

### 2.4. Electrocardiography Recording and Analysis (ECG)

The electrocardiogram (ECG) signal was measured by electrodes connecting to the data acquisition system, Power Labs (AD Instruments, Bella Vista, NSW, Australia). The ECG signal capture was accomplished with platinum electrodes inserted subcutaneously in three limbs and connected to a custom-built ECG amplifier and measured for about 5 min at 2 MHz.

### 2.5. Echocardiographic Analysis

The animals were anesthetized with 1.5% isoflurane in 95% oxygen and 5% carbon dioxide, and the hair of the chest was removed using a depilatory cream before the examination. The echocardiographic assessment was performed by a high-resolution ultrasound imaging system (VINNO 6, Vinno Corporation, Suzhou, China) in vivo heart function, and chamber dimensions were measured using a 23 MHz frequency transducer. M-mode recordings were obtained from the parasternal short-axis views. The internal dimensions of ejection fraction (EF), fractional shortening (FS), stroke volume (SV), cardiac output (CO), left ventricular posterior wall thickness at end-systole (LVPWs), left ventricular internal dimension at end-systole (LVIDs), interventricular septal thickness at end-systole (IVSs), and interventricular septal thickness at end-diastole (IVSd) were measured and recorded. Echocardiography measured the average of at least five readings was calculated.

### 2.6. Histological Analysis

Separated left ventricle and thoracic aorta tissues were placed in 0.01 M phosphate-buffered saline and 10% formalin and fixed for 24 h. Fixed tissues were embedded in paraffin, separated into thin sections of 3–6 μm thickness, and attached to the slide. The slides were stained with hematoxylin and eosin (H&E), Picrosirius Red Stain Kit (Picrosirius Red stain, Polysciences, Inc., Warrington, PA, USA), and Masson’s trichrome stain Kit (Masson Trichrome stain, BBC Biochemical, Mt Vernon, WA, USA) for histopathological comparisons and determined by the light microscopy (EVOSTM M5000, Thermo fisher scientific, Bothell, WA, USA).

### 2.7. Changes in Relaxation Response of Thoracic Aorta to Acetylcholine

The thoracic aorta was rapidly removed and immersed in ice-cold Krebs’s solution (pH 7.4, in mM:118.0 NaCl, 4.7 KCl, 25.0 NaHCO_3_, 10.0 glucose, 1.5 CaCl_2_, 1.1 MgSO_4_, and 1.2 KH_2_PO_4_), removed connective tissue, and cut into approximately 2–3 mm wide rings. Aortic rings were prepared as reported previously [19]. After an equilibration period of 60 min in organ chambers containing 5 mL Krebs solution at 37 °C, the isometric tension changes were recorded via a transducer (Grass FT 03, Grass Instrument Co., Quincy, MA, USA) connected to a Grass Polygraph recording system (Model 7E, Grass Instrument Co., Quincy, MA, USA.). The relaxation responses of aortic rings precontracted by phenylephrine (1 μM) to various acetylcholine were performed.

### 2.8. Western Blot Analysis and Antibodies

The thoracic aorta tissue (30 μg protein) were resolved on 10% SDS-polyacrylamide gel electrophoresis (SDS-PAGE) and blotted onto polyvinylidene difluoride membranes. Blots were then washed with TBS-T (10 mM Tris-HCl, 150 mM NaCl, 0.05% Tween-20) and blocked with 5% BSA or nonfat milk powder in 1X TBS-T for 2 h and incubated with the appropriate primary antibodies (HMGB1, TLR4, MyD88, NFκB, TNF-a, IL-6, RAGE, and ERK1/2) overnight at 4 °C. Primary antibody high mobility group box 1 (HMGB1, SC-74085), toll-like receptor 4 (TLR4, SC-293072), myeloid differentiation primary response 88 (MyD88, SC-136970), nuclear factor kappa-light-chain-enhancer of activated B cells (NFκB p65, SC-8008), tumor necrosis factor alpha (TNF-α, SC-1348), interleukin-6 (IL-6, SC-28343), receptor for advanced glycation end products (RAGE, SC-8230), extracellular signal-regulated kinases (ERK1/2, SC-514302), and β-actin were purchased from Santa Cruz Biotechnology (Dallas, TX, USA). The membrane was then washed with TBS-T and incubated with secondary antibodies conjugated to horseradish peroxidase (Bethyl Laboratories, Inc., Montgomery, TX, USA) for 2 h. Protein expression levels were determined membranes using the Chemi-doc image analyzer (iBright FL100, ThermoFisher Scientific, Waltham, MA, USA). Densitometry analysis of protein levels was conducted with the ImageJ program (National Institutes of Health (NIH), Bethesda, MD, USA).

### 2.9. Assessment of Cardiac Injury Biomarkers in Plasma

The plasma levels of atrial natriuretic peptide (ANP) were measured by radioimmunoassay as reported previously [20] and Lactate Dehydrogenase (LDH) were measured using an automated clinical chemistry analyzer (FUJI DRI-CHEM NX700, FUJIFILM Corporation, Tokyo, Japan).

### 2.10. Statistical Analyses

All experiments were repeated at least three times, and statistical analyses were performed by t-test. Results were expressed as mean ± standard error. The statistically significant difference between the group means was determined using the Student’s *t*-test. *p* < 0.05 was considered a statistically significant difference.

## 3. Results

### 3.1. HPLC Chromatograms of Quercetin-3-O-Glucuronide from HP1 Extract

HPLC analysis for HP1 from the method above providing Quercetin-3-O-glucuronide peak at 25.2 min, as shown in Figure 1. As a result, 1 g HP1 contain 14.13 ± 2.83 mg of Quercetin-3-O-glucuronide (Figure 1).

### 3.2. Effect of HP1 on Blood Pressure after Continuous Administration

All groups tested blood pressure four weeks later and compared according to HP1 treatment status. As shown in Table 1, blood pressure in the hypertensive rats was significantly higher than the WKY rat. On the other hand, the HP1-treated hypertensive rats had a significantly lower systolic blood pressure than the non-HP1 treated rats (HP1, 151.23 ± 3.5 mmHg vs. SHR 195.79 ± 4.06 mmHg) (Table 1).

### 3.3. Effects of HP1 on the ECG Changes in SHRs

To investigate the electrocardiographic parameter associated with HP1, we were examined the electrocardiography (ECG) changes. HP1-treated in hypertensive rats improved ECG parameters (a and b). The heart rate (Figure 2(Ba)) and QRS interval (Figure 2(Bb)) were significantly increased in hypertensive rats compared with the WKY. However, the RR interval decreased in hypertensive rats compared with the WKY (Figure 2(Bc)). The HP1-treated hypertensive rat had a significantly improved electrocardiographic parameter when compared with hypertensive rats.

### 3.4. Effect of HP1 on LV Remodeling and LV Function

To further evaluate whether HP1 affects cardiac function and wall thickness in hypertensive rats, we performed echocardiography. Figure 3A is a representative M-mode image, and the change in LV function and LV remodeling were compared between groups with echocardiogram parameters (Figure 3A). The ejection fraction (EF) and fractional shortening (FS) significantly impaired LV dysfunction in hypertensive rats compared with the WKY. On the other hand, HP1-treated hypertensive rats recovered LV dysfunction compared with the hypertensive rats (Figure 3(Ba–d)). Furthermore, hypertensive rats showed a significant increase in left ventricular posterior wall thickness at end-systole (LVPWs), interventricular septal thickness at end-systole (IVSs), and interventricular septal thickness at end-diastole (IVSd) (Figure 3(Be,g,h)), there was a significant decrease in the left ventricular internal dimension at end-systole (LVIDs) with in hypertensive rats (Figure 3(Bf)). On the other hand, LV remodeling and dysfunction caused by hypertension were suppressed with HP1 treatment.

### 3.5. Effect of HP1 on Cardiac Hypertrophy and Fibrosis in SHRs

In order to investigate the effects of treatment from HP1 administration on cardiac hypertrophy in hypertensive rats, the heart was measured by dividing it into four separate chambers. Hypertension is known to induce remarkable hypertrophy of cardiac. As shown in Table 2, the heart weight of the hypertensive rats was significantly increased in left atrial (LA), right atrial (RA), left ventricle (LV), right ventricle (RA), septa (Sep), and whole heart (WH) compared with WKY (Table 2). In addition, we found that the total WH weight/body weight ratio of hypertensive rats significantly higher than that of WKY (Figure 4A,B). On the other hand, the HP1-treated hypertensive rat had a significantly improved cardiac hypertrophy compared to hypertensive rats. To determine the protective effect of HP1 on cardiac fibrosis, histological analysis was performed (Figure 4C,D). Representative microscopic photographs in the LV were stained by Picrosirius Red staining (collagen fibers stained bright red) and Masson Trichrome staining (collagen fibers stained blue). As shown in Figure 4C,D, the fibrotic area was increased in hypertensive rat, while this was significantly attenuated by HP1. These results suggest that HP1 administration ameliorates cardiac hypertrophy and fibrosis in hypertensive rats.

### 3.6. Effects of HP1 on Vascular Remodeling in SHRs

The effects of improving blood vessel remodeling of HP1 in hypertensive rats were confirmed using the thoracic artery. The amount of vascular relaxation, blood vessel staining, and protein expression was measured. As a result, the effect of HP1 on vascular reactivity was tested on isolated aortic rings at hypertensive rats, endothelium-dependent relaxation to acetylcholine was significantly attenuated in hypertensive rats. It improved in HP1 groups compared with hypertensive rats (Figure 5A,B). Furthermore, the hypertension parameter ANP and LDH secretion in plasma increased in hypertensive rats, while reduced by the treatment of HP1 (Figure 5C,D).

To evaluate the effect of HP1 on vascular remodeling of hypertensive rats, we determined the media thickness of rat thoracic aorta using the H&E staining (Figure 6(Aa)). As a result, the thoracic aortic tissues showed that hypertrophy occurred in vascular smooth muscle cells of hypertensive rats (Figure 6(Ac)). HP1 treatment inhibited thoracic aortic hypertrophy caused by hypertension (Figure 6(Aa)). We assessed the expression of high mobility group box 1 (HMGB1), toll-like receptor 4 (TLR4), myeloid differentiation primary response 88 (MyD88), nuclear factor kappa-light-chain-enhancer of activated B cells (NFκB), tumor necrosis factor alpha (TNF-α), interleukin-6 (IL-6), receptor for advanced glycation end products (RAGE), and extracellular signal-regulated kinases (ERK1/2) levels in thoracic aorta using Western blotting. As illustrated in Figure 6B,C, HMGB1, TLR4, MyD88, NFκB, TNF-α, IL-6, RAGE, and ERK1/2 expression was significantly decreased in HP1-treated rats compared with hypertensive rats (Figure 6B,C). These results showed that HP1 could improve vascular remodeling in the hypertensive rats by attenuating the HMGB1-TLR4-NFκB signaling pathway. Therefore, Figure 7 shows representative images schematic proposal for the mechanisms responsible for improving cardiac dysfunction by HP1-treatment (Figure 7).

## 4. Discussion

We demonstrated that anti-hypertensive and improving cardiovascular remodeling effect of *Vitis labrusca* leaves extract (HP1) in the spontaneously hypertensive rat (SHRs, hypertensive rats). Furthermore, our results confirmed that the HP1 plays a role in HMGB1-TLR4-NFκB signaling that prevents heart fibrosis and hypertensive, improves cardiovascular dysfunction.

According to various research reports, wine is reported to have beneficial protective effects through coronary artery vascular expansion, reduction of intrinsic synthesis, low density lipoprotein, low blood pressure, and antioxidants [21,22,23]. In addition, the grape skin extract of *Vitis labuska* is reported to have improved hypertension models [24]. As such, studies on wine, grapes, and resveratrol (the main component of grapes) have been varied, but studies on the leaves of *Vitis labrusca* have not been actively conducted. Previous studies found that *Vitis labrusca* leaves were applied to investigate antithrombotic activity in vitro and ex vivo, suggesting them as a potential agent for preventing cardiovascular diseases [18]. Hence, the present study was designed to investigate the effects of *Vitis labrusca* leaves on cardiovascular remodeling in spontaneously hypertensive rats. Hypertension is associated with physiological changes, including cardiovascular remodeling due to the myocardium, resulting in left ventricular hypertrophy [25]. This study was conducted using an animal model (SHR) similar to human congenital hypertension [26]. The SHR is a model utilized to understand the mechanisms of hypertension and effective strategies for its treatment [27]. The SHRs develop blood pressure increases at 6 to 7 weeks of age and reach a stable level of hypertension, and then progressive cardiac hypertrophy, myocardial fibrosis, and heart failure [28,29]. We used a recognized model of experimental hypertension, and systolic myocardial dysfunction appeared in hypertensive rats.

As a positive control, losartan is an angiotensin receptor blocker and is known to be mainly used in the treatment of hypertension [30,31], and Losartan has an effect of reducing blood pressure in SHR, the hypertension model used in this experiment. In another study, losartan treatment of hypertensive rats with an angiotensin-converting enzyme inhibitor or angiotensin II type 1 receptor antagonist completely prevented hypertension and concomitant left ventricular hypertrophy [32,33,34,35].

The substrate for arrhythmia in the cardiac of hypertensive rats was associated with markedly increased fibrosis. The enlargement of the left ventricle is widely regarded as an epidemiological risk indicator for hypertension [36,37], and it has been suggested that atrial enlargement in hypertension contributes to the increased incidence of arrhythmia [38,39]. In order to better understand the effect of HP1 on bradycardia, we examined changes in the electrocardiogram (Figure 2). The findings of the research showed a significant change in the QRS interval values by ECG in hypertensive rats, being significantly increased compared to those in the WKY, and this difference was improved in the HP1-treated hypertensive rats.

Hypertension causes cardiovascular dysfunction such as endothelial dysfunction, hypertrophic heart dysfunction, increased fibrosis, and inflammatory changes through pathophysiological changes in blood vessels and myocardium [40,41]. The primary pathological change in hypertension is LV hypertrophy [42], which is characterized by increased muscle cell size and fibrosis and vascular reconstruction [43]. Cardiac hypertrophy is considered the main target of hypertension treatment [44], and compensatory thickening of the left ventricular wall was present to normalize wall stress of hypertensive rats [45]. Previous studies have also shown that hypertension is remodeled due to increased blood vessel thickness, LV internal diameter and cross-sectional area [43,46,47]. Therefore, it was confirmed through cardiac echocardiography and Masson’s trichrome staining analysis that administration of HP1 weakened the myocardial hypertrophy and cardiac remodeling in the hypertension rats (Figure 3 and Figure 4). In this study, we found that cardiac fibrosis is induced in the thoracic aorta of hypertensive rats compared to the WKY. On the other hand, HP1 treatment alleviated fibrosis in left ventricular and thoracic aortic tissues. Additioanlly, the acetylcholine-induced aortic ring relaxation was significantly attenuated in the hypertensive rats compared with the normal rats (Figure 5). These results indicated that vascular endothelial dysfunction had occurred in the hypertensive rats [48]. These results indicate that HP1 exhibits anti-hypertensive effects mediated through improvement of endothelial function in hypertensive rats. Therefore, these results suggest that HP1 may play a potential role in cardiovascular remodeling, and HP1 treatment significantly improved cardiac dysfunction in hypertensive rats.

In many hypertension reports, it has been concluded that hypertension causes inflammatory transcription factors directly affecting the vascular [49]. HMGB1 is a lethal inflammatory mediator; studies have shown that HMGB1 participates in disseminated intravascular coagulation, severe acute pancreatitis, burns, hemorrhagic shock, rheumatoid arthritis, systemic lupus erythematosus, and other diseases [50,51,52]. The major receptors of HMGB1 contain TLR4 and RAGE [53], MyD88 is utilized by all TLRs and activates NFκB for the induction of inflammatory cytokine genes [54]. The HMGB1 activates the proinflammatory cytokines associated with the TLR4-MyD88-NFκB signaling pathway, which increases inflammatory transcription factor [55]. Therefore, our present reports have shown that HP1 decreases the inflammatory response by reducing HMGB1-TLR4-MyD88-NFκB signaling formation. These results suggest that HP1 attenuated the cardiovascular dysfunction by blocking inflammatory molecules and their transcription factors. However, it is necessary to acknowledge and resolve the two major limitations of current research. First, we attempted to demonstrate the TNFa/ IL6 expression in aortic rings and not the secreted levels in rat plasma, and cellular signaling to TNFa/IL6 released by HP1 treatment is only speculative. Second, further study of relevant signaling is needed because activation of NF-κB involves decomposition of IκB and nuclear translocation of NF-κB. More experiments, such as cardiomyocyte investigation, should be performed in further studies to confirm the effect of HP1 in improving cardiovascular dysfunction.

## 5. Conclusions

In conclusion, these data show that HP1 has a protective effect on cardiovascular dysfunction in hypertensive rats. In addition, our results confirmed that HP1 has an effect on preventing cardiac fibrosis and improving cardiovascular remodeling by modulating HMGB1-TLR4-NFκB signaling. These results suggest that HP1 might prove to be effective in cardiac hypertrophy leading to hypertensive rats. Thus, the protective role of the traditional herbal medicine HP1 may provide new insights into the development of therapeutic drugs on the development of hypertensive cardiovascular dysfunction.

## Figures and Tables

**Figure 1 nutrients-12-03096-f001:**
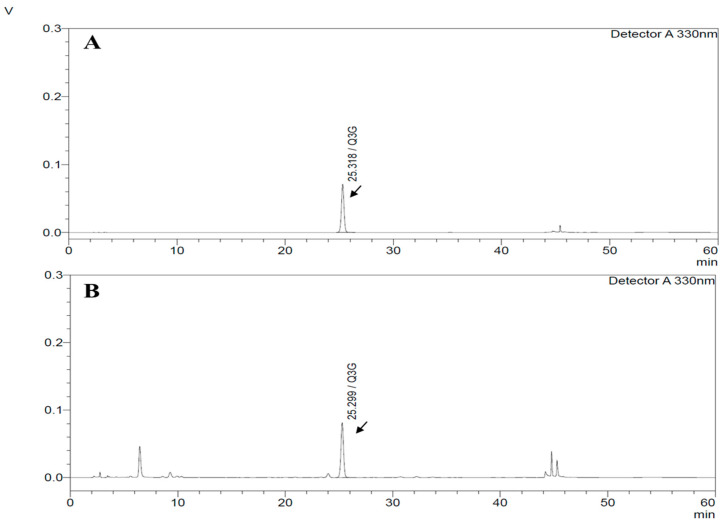
High-performance liquid chromatography (HPLC) chromatograms (detection at 330 nm) of Quercetin-3-O-glucuronide from *Vitis labrusca* leaves extract (HP1) extract. Standard (**A**) and HP1 extract (**B**).

**Figure 2 nutrients-12-03096-f002:**
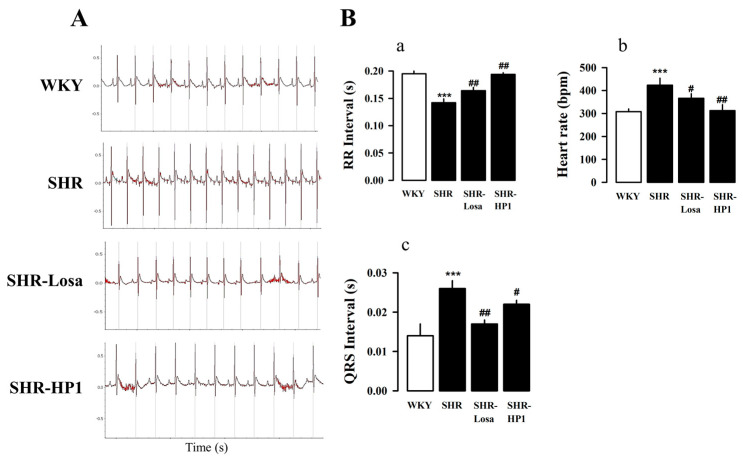
Effect of the electrocardiogram changes by HP1-treated in SHRs. Representative electrocardiogram (ECG) images of atrial and ventricular premature pulsation by HP1-treated in SHRs (**A**). The number of heart rate (**Ba**), RR interval (**Bb**), and QRS interval (**Bc**) for each rat was calculated from 5 min ECG recording. WKY, Wistar-Kyoto rat; SHR, spontaneously hypertensive rat; Losa, losartan; and HP1, *Vitis labrusca* leaves extract. Open bar, WKY; closed bar, SHRs. Data are shown as means ± standard error (*n* = 10~12 for each group). *** *p* < 0.001, compared with the WKY group; # *p* < 0.05 and ## *p* < 0.01 compared with the SHR group.

**Figure 3 nutrients-12-03096-f003:**
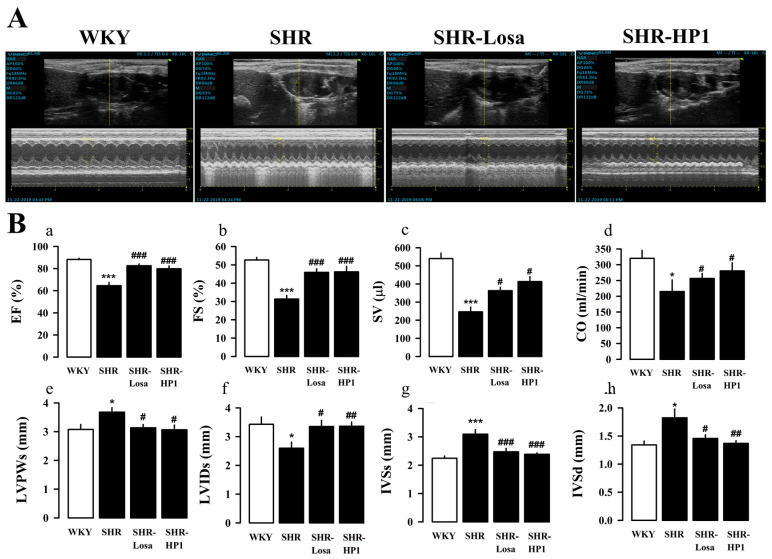
Effect of HP1 on left ventricle (LV) remodeling in SHRs. Representative echocardiography images (M-mode) for each group (**A**). Changes of ejection fraction (EF), fractional shortening (FS), stroke volume (SV), cardiac output (CO), left ventricular posterior wall thickness at end-systole (LVPWs), left ventricular internal dimension at end-systole (LVIDs), interventricular septal thickness at end-systole (IVSs), and interventricular septal thickness at end-diastole (IVSd) were measured echocardiography in each group (**Ba**–**h**). WKY, Wistar-Kyoto rat; SHR, spontaneously hypertensive rat; Losa, losartan; and HP1, *Vitis labrusca* leaves extract. Open bar, WKY; closed bar, SHRs. Data are shown as means ± standard error (*n* = 10~12 for each group). * *p* < 0.05 and *** *p* < 0.001, compared with the WKY group; # *p* < 0.05, ## *p* < 0.01, and ### *p* < 0.001 compared with the SHR group.

**Figure 4 nutrients-12-03096-f004:**
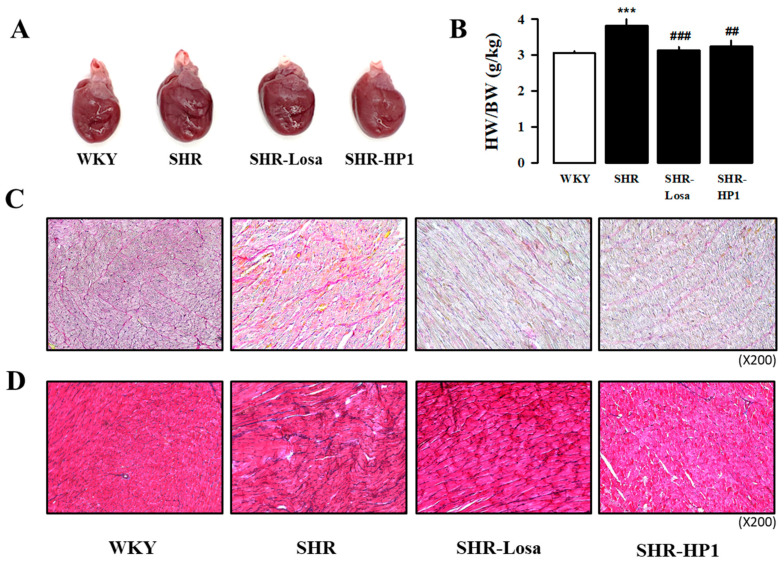
Effect of HP1 on cardiac hypertrophy and fibrosis. The HP1 treatment attenuated ventricular remodeling in SHR. The appearance of the whole hearts from each group (**A**). The numerical data to calculate the heart weight/body weight (HW/BW) ratio are shown (**B**, *n* = 10~12 for each group). The detailed anatomy of the changes in picrosirius red staining (**C**, upper panels) and Masson’s trichrome staining (**D**, *n* = 3 for each group) of ventricular tissue from each group. WKY, Wistar-Kyoto rat; SHR, spontaneously hypertensive rat; Losa, losartan; and HP1, *Vitis labrusca* leaves extract. Open bar, WKY; closed bar, SHRs. Data are shown as means ± standard error. *** *p* < 0.001, compared with the WKY group; ## *p* < 0.01, and ### *p* < 0.001 compared with the SHR group.

**Figure 5 nutrients-12-03096-f005:**
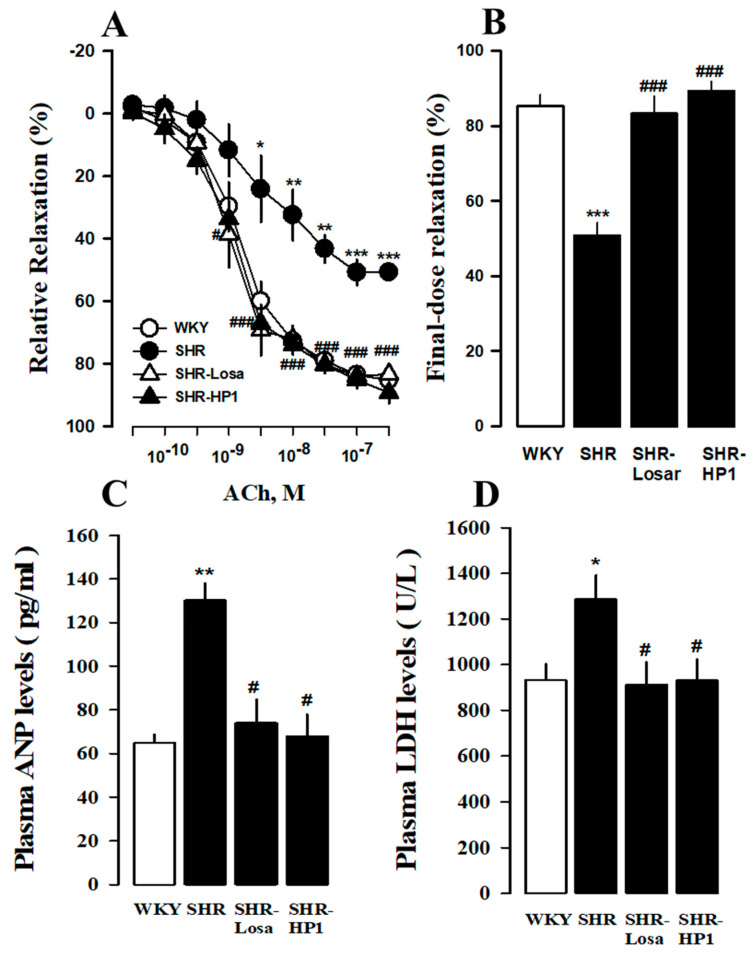
Effects of HP1 on isolated rings of thoracic aorta and hematological change in SHRs. Vascular relaxation by acetylcholine (Ach, **A**) and relaxation by acetylcholine final dose (**B**) in thoracic aortic rings. The levels of plasma atrial natriuretic peptide (ANP, **C**) and lactate dehydrogenase (LDH, **D**) of HP-1-treated in SHRs. WKY, Wistar-Kyoto rat; SHR, spontaneously hypertensive rat; Losa, losartan; and HP1, *Vitis labrusca* leaves extract. Open bar, WKY; closed bar, SHRs. The values represent the means ± standard error (*n* = 5 for each group). * *p* < 0.05, ** *p* < 0.01, *** *p* < 0.001 vs. WKY; # *p* < 0.05, and ### *p* < 0.001 vs. SHRs.

**Figure 6 nutrients-12-03096-f006:**
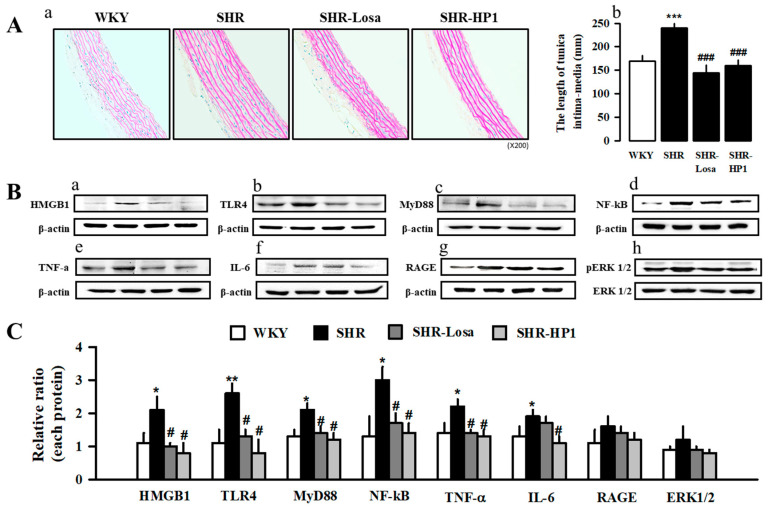
Effect of HP1 on thoracic aorta remodeling in SHRs. Histological changes in the thoracic aorta after HP1 administration to SHR. Representative images of hematoxylin and eosin (H&E) staining in the thoracic aorta of HP-1-treated in SHRs (**Aa**). Thoracic aorta ring media thickness in the indicated treatment groups (**Ab**). Effects of HP1 on the expression levels of HMGB1-TLR4-NFκB signaling in the thoracic aorta from SHRs. The protein expressions of HMGB1, TLR4, MyD88, NFκB (p65), TNF-a, IL-6, RAGE, and ERK1/2 were measured by Western blot analysis in thoracic aorta from each group (**Ba**–**h**). The bar represents the relative protein quantification of proteins based on β-actin (**C**). WKY, Wistar-Kyoto rat; SHR, spontaneously hypertensive rat; Losa, losartan; and HP1, *Vitis labrusca* leaves extract; HMGB1, high mobility group box 1; TLR4, toll-like receptor 4; MyD88, myeloid differentiation primary response 88; NFκB, nuclear factor kappa-light-chain-enhancer of activated B cells; TNF-α, tumor necrosis factor alpha; IL-6, interleukin-6; RAGE, receptor for advanced glycation end products; and ERK1/2, extracellular signal-regulated kinases. The values represent the means ± standard error (*n* = 4 for each group). * *p* < 0.05, ** *p* < 0.01, *** *p* < 0.001 vs. WKY; # *p* < 0.05, and ### *p* < 0.001 vs. SHRs.

**Figure 7 nutrients-12-03096-f007:**
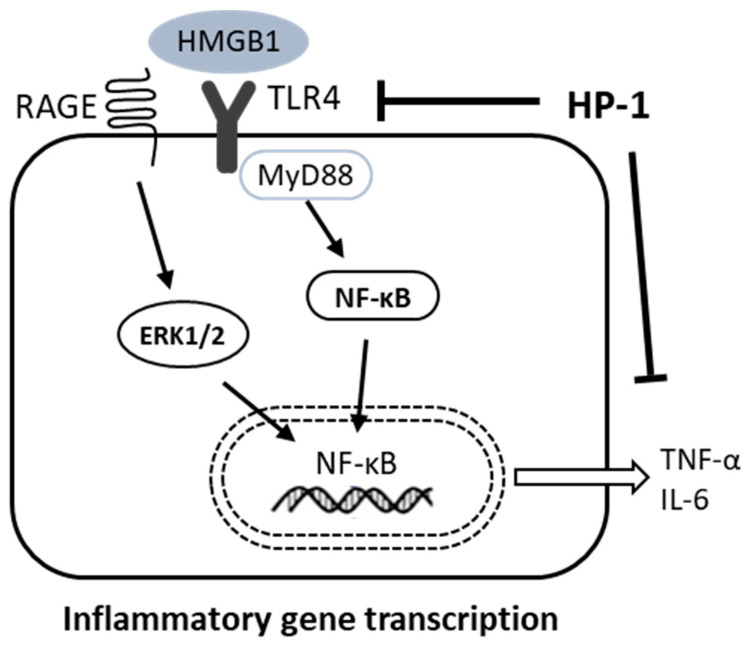
Schematic diagram showing that HP1 inhibits cardiac dysfunction in SHRs through intracellular HMGB1-TLR4-NFκB signaling suppression. SHR, spontaneously hypertensive rat; HP1, *Vitis labrusca* leaves extract; HMGB1, high mobility group box 1; TLR4, toll-like receptor 4; MyD88, myeloid differentiation primary response 88; NFκB, nuclear factor kappa-light-chain-enhancer of activated B cells; TNF-α, tumor necrosis factor alpha; IL-6, interleukin-6; RAGE, receptor for advanced glycation end products; and ERK1/2, extracellular signal-regulated kinases.

**Table 1 nutrients-12-03096-t001:** Analysis of cardiac function characteristics by HP1-treated in SHRs.

	SBP			DBP			MAP			HR			Flow			Volume		
WKY	106.31	±	3.82	94.68	±	5.00	100.03	±	4.61	353.30	±	8.24	2.23	±	0.39	3.26	±	1.29
SHR	192.79	±	4.06 ^***^	167.18	±	5.53 ^***^	170.51	±	5.37 ^***^	477.03	±	8.49 ^**^	5.08	±	1.08 *	10.01	±	1.90 ^**^
SHR-Losa	147.93	±	3.09 ^##^	135.03	±	4.11 ^##^	129.12	±	3.60 ^##^	377.31	±	9.82 ^#^	4.79	±	1.72	5.44	±	1.80 ^#^
SHR-HP1	151.23	±	3.50 ^##^	120.86	±	7.35 ^##^	152.98	±	6.82 ^##^	365.61	±	9.30 ^#^	3.87	±	0.81	7.72	±	1.77 ^#^

WKY, Wistar-Kyoto rat; SHR, spontaneously hypertensive rat; Losa, losartan; HP1, *Vitis labrusca* leaves extract; SBP, systolic blood pressure; DBP, diastolic blood pressure; MAP, mean arterial pressure; and HR, heart rate. Values are expressed as the means ± standard error (*n* = 10~12 for each group). * *p* < 0.05, ** *p* < 0.01 and *** *p* < 0.001 compared with the WKY group; # *p* < 0.05, and ## *p* < 0.01 compared with the SHR group.

**Table 2 nutrients-12-03096-t002:** Effect of HP1 on weight changes in heart chamber by SHRs.

	LA			RA			LV			RV			Sep			WH		
WKY	15	±	0.001	19	±	0.002	388	±	0.021	128	±	0.009	146	±	0.012	699	±	0.015
SHR	17	±	0.001 ^*^	23	±	0.001 ^*^	515	±	0.01 ^**^	153	±	0.006 ^**^	177	±	0.01 ^**^	882	±	0.005 ^**^
SHR-Losa	14	±	0.001	18	±	0.001	415	±	0.012 ^#^	146	±	0.004	153	±	0.001 ^#^	746	±	0.016 ^#^
SHR-HP1	15	±	0.001	20	±	0.001	429	±	0.031 ^#^	149	±	0.010	170	±	0.01 ^#^	782	±	0.032 ^#^

WKY, Wistar-Kyoto rat; SHR, spontaneously hypertensive rat; Losa, losartan; HP1, *Vitis labrusca* leaves extract; LA, left atrium; RA, right atrium; LV, left ventricle; RV, right ventricle; Sep, septa; and WH, Whole Heart. Values are expressed as the means ± standard error (*n* = 10~12 for each group). * *p* < 0.05 and ** *p* < 0.01 compared with the WKY group; # *p* < 0.05 compared with the SHR group.

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
