# Peer review of "Protective Effect of Vitis labrusca Leaves Extract on Cardiovascular Dysfunction through HMGB1-TLR4-NFκB Signaling in Spontaneously Hypertensive Rats"

_nutrients, 2020, doi:10.3390/nu12103096_

Round 1

Reviewer 1 Report

This manuscript reports the protective effect of Vitis labrusca leaves extract on cardiovascular dysfunction in spontaneously hypertensive rats. Authors found that Vitis labrusca leaves extract (HP1) improved the cardiovascular remodeling in hypertensive rats, and the mechanisms may be related to the suppressive effect of HP1 on HMGB1-TLR4-NFκB signaling in the cardiovascular system of hypertensive rats.

This study is interesting. However, there are some questions needed to be addressed.

1. Authors analyzed the effects of HP1 by oral administration of HP1 (100 mg/kg/daily) to SHR for 4 weeks. Why did authors use the dosage of 100 mg/kg HP1? In addition, dose response is necessary to explain the biological function of compounds/drugs. Various amounts of HP1 should be given to SHRs to see if the protective effects display a dose-dependent manner.

2. NF-κB is a transcription factor, consisting of p65/p50 or p52 protein. Authors should describe the type (anti-p65 or others) of primary antibody applied to detect NF-κB in Western blot. In addition, the activation of NF-κB is associated with the degradation of IκB and the nuclear translocation of NF-κB. The amount of p65 in nuclear extract should be determined.

3. Soares De Moura et al (2002) have reported the antihypertensive, vasodilator and antioxidant effects of a vinifera grape skin extract in the Journal of Pharmacy and Pharmacology. Authors should compare their data with authors' data in Discussion section.

Reviewer 2 Report

In this interesting paper, Kim et al. explore the therapeutic effectiveness of Vitis labrusca leave extract (HP1) in an in vivo model of cardiovascular remodeling in spontaneously hypertensive rats (SHR).

The authors observed distinct protective effects on hearts of SHR after treatment with losartan and/or HP1 and they consecutively explored the underlying signaling pathways, thereby identifying the involvement of HMGB1-TLR4-NFκB signaling pathway.

The experimental approaches are sound and state of the art. In their results section and discussion, the authors delineate a novel mechanism, outlining how HP1 ameliorates heart function in SHR rats.

However, there are considerable flaws in the experimental design and a number of major issues that should be addressed, as outlined in detail below.

Major Comment:

1) In none of the figures the number of animals/tissues was indicated. Therefore I would assume the animal/tissues numbers were n=3? How could the authors get P-values with significance of P<0.001?

2) Why did the authors measure TNFa, IL6 expression in aortic rings and not the secreted levels in rat blood / serum?

3) In figure 7, the representative blots for TLR4, TNFa, IL6, RAGE and pERK do not fit with the shown bar graphs.

4) The conclusion number 7 the authors made was not supported by the data the authors provided. The signalling from HP1 to TNFa/IL6 release is speculative. To analyse the cellular signaling the authors should use cardiomycotes as target cells.

Minor point:

Did the authors test also a group for the solvent control for HP1?

Reviewer 3 Report

Kim et al. in the present research study entitled “Protective effect of Vitis labrusca leaves extract on cardiovascular dysfunction through HMGB1-TLR4-NFκB signaling in spontaneously hypertensive rats” investigated the protective effects of Vitis labrusca leaves extract (HP1) on cardiovascular remodeling in spontaneously hypertensive rats. The authors found that treatment with HP1 significantly reduced systolic blood pressure, improved electrocardiogram and echocardiogram compared to controls. They also reported a decrease in the fibrotic area and reduction in NFκB signaling.  The study is well-designed and the “Materials and Methods” section is adequately detailed. However, I have the following comments:

  1. The manuscript has numerous grammatical errors, and needs critical language editing.
  2. The authors investigated endothelium-dependent vascular relaxation using Ach. As SHRs have thicker aortic medial layer compared to WKY rats, did authors observe the effects of HP1 treatment on endothelium-independent vascular relaxation employing SNP?
  3. I did not see any fibrosis in Fig. 4D, and there is no specific staining in Fig. 6B. Can the authors provide high magnification images for both figures?
  4. 6B and 6C, Western blot representative images do not match with protein expression data in Fig. 6C (e.g., TLR4 and TNFa).
  5. In table 1, I am surprised to see such a high diastolic blood pressure in the SHR group. In addition, the authors need to mention the number of rats used in each group in the figure legends. This applies to every figure and each panel. They should include n number of times each experiment was performed.
  6. 4C, in the figure legend, it is written that representative images of H & E staining are shown, however, text indicated it as Picrosirius Red staining. Please correct.

Round 2

Reviewer 1 Report

Authors concluded that HP1-improved cardiovascular remodeling in hypertensive rats may be related to the suppressive effect of HP1 on HMGB1-TLR4-NFκB signaling in the cardiovascular system of hypertensive rats. Because the activation of NF-κB is associated with the degradation of IκB and the nuclear translocation of NF-κB. I suggest that the amount of p65 in nuclear extract or the amount of IkB in the cytoplasm should be analyzed by Western blot.

Author Response

Comment: Authors concluded that HP1-improved cardiovascular remodeling in hypertensive rats may be related to the suppressive effect of HP1 on HMGB1-TLR4-NFκB signaling in the cardiovascular system of hypertensive rats. Because the activation of NF-κB is associated with the degradation of IκB and the nuclear translocation of NF-κB. I suggest that the amount of p65 in nuclear extract or the amount of IkB in the cytoplasm should be analyzed by Western blot.

Response: I really thank you very much for your invaluable suggestions and comments. As the reviewer suggested, activation of NF-κB is associated with the degradation of IκB and the nuclear translocation of NF-κB, but we've only checked at the NF-κB. In the discussion about the limitations of these studies, we have mentioned should be performed in further studies (page 12, section 4).

Thanks for reviewing the manuscript submitted, and we really thank Reviewer #1 very much indeed.

Reviewer 2 Report

The authors have adequately addressed most, but not all, of the comments raised and in my opinion have significantly improved the manuscript.

Author Response

Comment: The authors have adequately addressed most, but not all, of the comments raised and in my opinion have significantly improved the manuscript.

Response: Thanks for reviewing the manuscript submitted, and we really thank Reviewer #2 very much indeed.

Reviewer 3 Report

The authors have significantly improved the manuscript, and answered most of the raised concerns satisfactorily. However, HMGB1 immunofluorescence data shown in Fig. 6 is still debatable. It is advised that they may either remove that immunofluorescence data as Western blot of HMGB1 is sufficient to demonstrate change in its expression or perform immunofluorescence analysis again.

Author Response

Comment: The authors have significantly improved the manuscript, and answered most of the raised concerns satisfactorily. However, HMGB1 immunofluorescence data shown in Fig. 6 is still debatable. It is advised that they may either remove that immunofluorescence data as Western blot of HMGB1 is sufficient to demonstrate change in its expression or perform immunofluorescence analysis again.

Response: We have removed the Fig. 6Ab as suggested, and we have revised the text in the Results and Figure legends. Thanks for reviewing the manuscript submitted, and we really thank Reviewer #3 very much indeed.